# A public context with higher minority stress for LGBTQ* couples decreases the enjoyment of public displays of affection

Michelle Stammwitz⬡, Janet Wessler *⬡

Department of Social Psychology, Saarland University, Saarbrücken, Germany

⬡ These authors contributed equally to this work.
* janet.wessler@uni-saarland.de

**Data Availability Statement:** All materials, including survey codes, data, syntaxes, and questionnaires, can be found on the OSF: https://osf.io/hcv68/?view_only=2f23384a6ae94dfcb7aa731fc88e95b4.

## Abstract

This research investigated whether LGBTQ* minority stress and public displays of affection (PDA; e.g., kissing, hugging) among LGBTQ* couples are context-sensitive. We expected that (a) LQBTQ* minority stress would be more prevalent in a harmful (i.e., city center) versus a less harmful (i.e., university campus) context, and (b) PDA would be reduced for LGBTQ* couples in a harmful context. In three studies, LGBTQ* and Hetero/Cis students ($N_{Total}$ = 517) reported LGBTQ*-specific minority stress and PDA in the city and on campus. The city center was higher in minority stress than the campus in all studies. Also, LGBTQ* participants' PDA enjoyment was lower in the city than on campus (Studies 1 and 3). Minority stress mediated the context effect on PDA (Study 3). A qualitative analysis illuminated the harmful versus protective natures of public contexts. We conclude that a protective context can powerfully promote healthy LGBTQ* relationship behavior.

## Introduction

"We must have kissed or something because these guys came after us" (Melania Geymonat as cited in [1]). This statement by a victim of physical harassment describes an incident in which a lesbian couple was attacked on a bus in London after engaging in a public display of affection (PDA). Although Western societies celebrate progress in the rights of lesbian, gay, bisexual, transgender, and queer/questioning (LGBTQ*) people, this incident emphasizes that LGBTQ* couples still risk violent reactions when displaying PDA. Is this true in every public context though?

Engaging in PDA means disclosing one's sexual (minority) identity. As (visible) members of a minority group, LGBTQ* individuals and couples are generally more likely to experience minority stress than heterosexual/cisgender (Hetero/Cis) individuals. These minority stress experiences (e.g., discrimination or physical violence) create a hostile social environment for LGBTQ* individuals and couples. These experiences might also explain why some LGBTQ* couples hesitate to engage in PDA. Indeed, minority stress was negatively associated with PDA in previous research [2, 3]. However, this research did not investigate whether both minority stress and PDA are sensitive to different contexts.

**Funding:** The authors received no specific funding for this work.

**Competing interests:** The authors have declared that no competing interests exist.

The current research is aimed at investigating whether (the stereotypical) perception of LGBTQ* minority stress changes across different public contexts and whether this change influences the PDA of LGBTQ* couples.

## Minority stress

According to Meyer's minority stress theory [4], members of a minority group are (more) likely to face minority stress. An "underlying assumption" of the theory is that minority stress is both "unique" and "additive to general stressors that are experienced by all people, and therefore, stigmatized people require an adaptation effort above that required of similar others who are not stigmatized" [4, p. 676]. Meyer (2003) also suggested a distinction between distal and proximal stressors. Distal stressors are typically incidents that are objectively harmful and tied to the reaction of the outside world. They include a variety of behaviors, such as physical attacks, verbal (harassment), and para-verbal societal marginalization (e.g., a disapproving look or spitting on the floor). Proximal stressors are more subjective as they include individual perceptions and internalized stigmatization. Our research focuses mainly on distal stressors. However, Meyer pointed out that the proximal-distal distinction may be viewed as falling along a continuum with distal factors laying the groundwork for proximal (personal) stress.

Regardless of any differentiation, the experience of minority stress typically has a negative impact on several dimensions of personal well-being and health [5–8]. Minority stress seems to increase the risk of psychopathology and appears to have an overall negative effect on mental health [9]. For example, minority stress (e.g., perceived discrimination based on sexual orientation) is associated with higher emotional distress [10] and higher levels of depressive symptoms and suicidal thoughts among LGBTQ* youth [11]. For a review of minority stress as a social determinant of health, see [12].

So far, research investigating the effects of minority stressors has primarily focused on the individual, but it is easy to imagine that minority stress could also affect LGBTQ* couples. Indeed, a meta-analysis found that negative attitudes, judgments, or behaviors toward LGBTQ* individuals and couples were inversely associated with relationship functioning [13]. Similarly, another meta-analysis reported that LGBTQ* minority stressors were negatively correlated with relationship well-being [14].

## Physical affection

Kissing, hugging, holding hands, and physical affection in general are positively related to relationship outcomes for both heteronormative and sexual minority couples [3, 15, 16]. Physical displays of affection refer to "any touch intended to arouse feelings of love in the giver and/or the recipient," [13, p. 234]. Typically, physical displays of affection include seven types of behavior: holding hands, hugging, kissing on the lips, kissing on the face, caressing/stroking, cuddling/holding, and backrubs/massages [15]. These types of behavior are thought of as non-sexual, romantic, and not as a precursor to intercourse.

Acts of physical affection are positively related to relationship satisfaction [15]. Moreover, hugs from one's partner are associated with an increase in the release of the hormone oxytocin, which helps couples form lasting relationship bonds [17]. Research has also suggested that physical affection is good not only for one's relationship but also for one's own well-being [17]. For example, physical affection seems to reduce a person's reactivity to stressful events [18].

In addition to its relevance on the levels of the health of relationships and individuals, the PDA of LGBTQ* couples might also be relevant on a societal level. In countries with LGBTQ* friendly legislation (e.g., equal marriage rights), LGBTQ* couples and families are an

increasingly visible part of everyday life. As such, LGBTQ* couples' PDA can help reduce LGBTQ*-related prejudice [19].

However strong these positive implications of physical affection and PDA might be, minority stress may discourage same-sex-oriented individuals from expressing affection in public or from enjoying displaying affection toward their partner. Even heterosexual individuals who support equal rights (e.g., gay marriage) are less willing to accept PDA shown by LGBTQ* couples compared with heteronormative couples [20].

Brady [2] investigated the frequency of PDA and three minority stressors: perceived danger, fear of heterosexism, and internalized homonegativity. Perceived danger referred to the fear of being physically attacked because of one's LGBTQ* identity (e.g., after engaging in PDA with a same-sex partner), whereas fear of heterosexism referred to physically nonviolent acts of discrimination. The third stressor, internalized homonegativity, referred to internalized negative attitudes about LGBTQ* individuals. All three stressors were negatively correlated with PDA. Hocker et al. [21] investigated a dyadic sample of same-gender couples and also found a negative effect of minority stress on PDA. Kent and El-Alayli [3] compared female same-sex couples with different-sex couples regarding the frequency with which they engaged in private and public displays of affection. They found a negative correlation between perceived marginalization and the frequency of PDA. Perceived marginalization mediated the effect of type of relationship on PDA for female couples. Moreover, both public and private displays of affection predicted higher relationship satisfaction, whereas perceived marginalization was associated with lower relationship satisfaction [3].

These findings demonstrate that minority stress has far-reaching consequences. Not only does it affect everyday life by making the choice to engage in PDA a daunting one. It also seems to decrease PDA, which is associated with lower long-term individual and relationship satisfaction.

## Context

Even though the findings concerning the PDA of LGBTQ* couples seem discouraging at times, there is no such thing as *the* public context. It seems intuitive that the degree of perceived minority stress varies across public contexts (e.g., name-calling in a gay bar vs. a sports bar) as much as the degree of PDA (e.g., kissing in a café vs. a church).

Previous research on stereotyping has shown that variation can occur across different contexts [22, 23]. In an implicit association test participants showed racial bias in a safe context (e.g., a family barbecue or a church) but not so much in a dangerous context (e.g., a gang incident or a street corner) [22]. The authors also emphasized the advantage of a within-subjects design as it revealed that different contexts could activate different attitudes in the same person.

Also, previous research found that LGBTQ* experiences are inherently connected to the social context in which one lives (e.g., one's neighborhood) as it can provide or deny access to LGBTQ*-affirmative resources on a community level (e.g., LGBTQ*-supportive places, pride marches, LGBTQ* couples holding hands in public [24]). Living in a more protective context can be associated with more social support and lower internalized heterosexism. Also, LGBTQ* women who moved from a context with low access to LGBTQ*-affirmative resources to a context with high access reported less minority stress and more outness than women who remained in a more heterosexist context [24]. In this sense, identifying and creating safe contexts is highly relevant to the personal and relationship health of LGBQT* individuals.

## The current research

Putting together the potential risks of minority stress with the potential benefits of displays of affection, PDA seems to be a tug of war for LGBTQ* couples. We propose that this tug of war

could be substantially influenced by (changing) the public context. To the best of our knowledge, this research is the first to investigate the context-sensitivity of minority stress and PDA and, hence, to explore the protective nature of the social context for LGBTQ* individuals and couples.

We propose that a city center is a public context with higher minority stress compared with a university campus. We argue that a campus is usually populated by individuals who are young and have an above-average education, which are two variables associated with more favorable attitudes toward homosexuality [25]. This notion is supported by findings that younger generations are politically more liberal. For example, cohort effects explained most of the increased support for equal marriage rights in California [26]. Previous research on campus climate also showed that universities may lack the level of LGBTQ* friendliness needed to be a truly welcoming and inclusive space for minorities [27] but that the situation has been improving [28]. For example, LGBTQ* students who graduated more recently reported more positive perceptions than earlier graduates [28].

A city center, however, is socially more diverse concerning age and political attitudes, which might be associated with less favorable attitudes toward minorities. Holding more conservative attitudes, for example, is associated with anti-gay attitudes [29]. Also, older adults seem to be less approving of LGBTQ* visibility [30, 31]. These factors might spawn concerns about safety and being scrutinized as a couple, two stressors reported by LGBTQ* couples when talking about being "out in public" [32, p. 465]. Hence, a city center may be seen as a harmful social context compared with a university campus.

In accordance with these assumptions, we aimed to explore (a) whether minority stress differs among public social contexts, (b) whether this context-specific difference in minority stress translates into couple behavior (PDA), and finally (c) what the potential protective value of the social context might be. These research questions directly add to research on social contexts as protective or harmful environments for LGBTQ* individuals. Answering these research questions will aid a better understanding of the potential of social contexts to be a tool for promoting healthy relationship behavior in LGBTQ* couples.

First, we hypothesized that LQBTQ* minority stress would be more prevalent in a harmful (i.e., city center) versus less harmful (i.e., university campus) public context (Hypothesis 1). Second, we expected that the PDA of LGBTQ* individuals would be reduced compared with heteronormative individuals (Hypothesis 2). Third, we expected that a harmful versus a less harmful context would be associated with reduced (enjoyment of) PDA among LGBTQ* couples, whereas heteronormative couples' reports of PDA would not be sensitive to the public context (Hypothesis 3).

To test these hypotheses, we conducted three online survey studies with sexual minority and majority participants in two different countries. Participants answered questions about their sexual and gender identity, perceived minority stress, and public displays of affection with a (hypothetical) partner both in the city and on campus. Study 1 ($N = 78$) was conducted in Ireland, whereas Study 2 ($N = 168$) and Study 3 ($N = 268$) were conducted in Germany. Studies 1 and 2 were aimed at establishing and replicating context effects of minority stress and PDA, respectively. Study 3 was designed to additionally test a mediating role of minority stress on PDA and to qualitatively examine the effects.

**Data analysis.** The data were analyzed using IBM SPSS 24 for quantitative analyses and MAXQDA 2020 [33] for qualitative analyses.

## Study 1

In Study 1, we aimed to establish the effect of different public contexts on perceived minority stress and at investigating the context-sensitivity of PDA depending on participants' sexual and gender identities. Study 1 was conducted in Limerick, Ireland in 2018.

## Ethics statement

The Faculty of Education and Health Sciences Research Ethics Committee of the University of Limerick approved the research (approval number: 2018_10_13_EHS) Consent was obtained in written form (online) and the data were analyzed anonymously.

## Method

The complete study materials for all the studies, including the survey codes, data, syntaxes, and questionnaires, can be found on the Open Science Framework (OSF): https://osf.io/hcv68/? view_only=2f23384a6ae94dfcb7aa731fc88e95b4. There were no conflicts of interest in conducting this research. This research was conducted ethically, the results are reported honestly, the submitted work is original and not (self-)plagiarized, and the authorship reflects the authors' individual contributions.

**Design.** Study 1 used a 2 (Context: Campus vs. City) x 2 (Sexual and gender identity: LGBTQ* vs. Hetero/Cis) mixed design with sexual and gender identity as a quasi-experimental between-participants factor and context as a within-participants factor. The dependent variables were minority stress and PDA enjoyment, which were rated separately for each context.

**Participants.** The survey was completed by $N = 78$ university students ($n_{LGBTQ^*} = 34$, $n_{Hetero/Cis} = 44$) who were between 18 and 60 years of age ($M_{age} = 20.97$, $SD_{age} = 5.74$). In this sample, $n = 50$ participants identified as female (64%), $n = 1$ participant identified as other/ diverse (1%), and $n = 27$ participants reported being in a committed relationship (35%). Table 1 shows the distribution of participants' self-reported sexual and gender identity in all studies.

**Independent variables.** *Sexual and gender identity.* We used sexual and gender identity as a quasi-experimental variable that separated our sample into LGBTQ* and Hetero/Cis participants. Participants who self-identified as lesbian, gay, bisexual, queer, pansexual, questioning,

**Table 1. Distribution of self-reported sexual and gender identification.**

|  | Study 1 | | Study 2 | | Study 3 | |
|---|---|---|---|---|---|---|
|  | **n** | **%** | **n** | **%** | **n** | **%** |
| Sexual identification |  |  |  |  |  |  |
| Heterosexual | 40 | 51.3 | 94 | 55.0 | 129 | 48.1 |
| Lesbian | 3 | 3.8 | 10 | 5.8 | 23 | 8.6 |
| Gay | 10 | 12.8 | 9 | 5.3 | 16 | 6.0 |
| Bisexual | 10 | 12.8 | 36 | 21.1 | 31 | 11.6 |
| Pansexual | 1 | 1.3 | 8 | 4.7 | 20 | 7.5 |
| Queer | 3 | 3.8 | 4 | 2.3 | 26 | 9.7 |
| Asexual | 3 | 3.8 | 3 | 1.8 | 8 | 3.0 |
| Other | 3 | 3.8 | 7 | 4.1 | 3 | 1.1 |
| Questioning | 5 | 6.4 | — | — | 12 | 4.5 |
| Gender identification |  |  |  |  |  |  |
| Cisgender | 69 | 88.5 | 152 | 88.9 | 215 | 80.2 |
| Transgender | 1 | 1.3 | 5 | 2.9 | 11 | 4.1 |
| Nonbinary* | 5 | 6.4 | 9 | 5.2 | 22 | 8.2 |
| Other | 2 | 2.6 | 3 | 1.8 | 7 | 2.6 |
| Questioning | — | — | — | — | 9 | 3.4 |
| *N* | 78 |  | 171 |  | 268 |  |

*Note.* All percentages refer to the whole sample. Percentages that do not add up to 100% are due to missing values. *Self-Identification as Genderqueer, Genderfluid, Gender Nonconforming, and Agender can also be found under the category of Nonbinary.

transgender, genderqueer or genderfluid were categorized as LGBTQ*. Participants who did not self-identify with either a sexual (e.g., lesbian) or gender (e.g., transgender) minority identity were categorized as Hetero/Cis. Participants who belonged to one or more sexual or gender minority groups were categorized as LGBTQ*. For example, a participant who reported being lesbian but identified with their assigned sex (cisgender) was categorized as LGBTQ*. The same applied for a participant who reported being attracted to the opposite gender (heterosexual) but did not identify with the gender they were assigned at birth (transgender).

*Context.* For context as an independent variable, we differentiated between the contexts of a university campus and a city center. We defined the campus as a university's openly accessible space (i.e., outside seating areas or cafés, not a lecture room). The city center was defined as openly accessible areas, such as the pedestrian zone, the shopping mile, or cafés. Participants rated minority stress and PDA for both contexts.

**Dependent variables.** Table 2 presents the means, standard deviations, and internal consistency scores (Cronbach's alpha) of the dependent variables.

*Minority stress.* For the purpose of these studies, we created a short minority stress scale (SMSS) that assessed LGBTQ* minority stress with four items. We used the following instructions:

> "We would like you to evaluate your social environment in terms of unjust behavior. Please give us your own opinion and remember there are no right or wrong answers. Please assess the situation [on the campus of your university/in the corresponding city center] from your personal point of view."

Two items assessed minority stress experiences and incidents concerning LGBTQ* individuals (i.e., "What do you think–How often have you heard other people [on the campus of your university/in the city center] make disparaging remarks about lesbian, gay, transgender, or other queer people?"; "What do you think–How often have you seen people give disparaging looks [on campus of your university/in the city center] that were meant for lesbian, gay, transgender, or other queer people because of their sexual or gender identity?"), and answers ranged from 1 (*never*) to 7 (*very often*).

Two further items assessed the perceived likelihood of discrimination and physical harassment of LGBTQ* individuals (i.e., "What do you think–How likely is it that an average LGBTQ* person will be the target of harassment, threat of violence, or physical attack [on the campus of your university/in the city center]?"; "What do you think–How likely is it that an average LGBTQ* person will be the target of discrimination or unfair treatment [on the campus of your university/in the city center]?"). Answers ranged from 1 (*not at all likely*) to 7 (*very*

**Table 2. Summary of correlations, reliabilities, means, and standard deviations for scores on scales for PDA and minority stress in Study 1.**

| Measure | *1* | *2* |
|---|---|---|
| 1. PDA Enjoyment Scale | .93 | —— |
| 2. Minority Stress (SMSS) | — .19 | .87 |
| | (.089) | |
| *M* | 5.10 | 3.33 |
| *SD* | 1.49 | 1.27 |

*Note*. SMSS = Short Minority Stress Scale. Reliabilities (Cronbach's alpha) are presented on the diagonal, *p*-value is presented in parentheses.

*likely*). Please note that the wording of the scale allowed participants to answer the questions regardless of their own sexual or gender identity.

*PDA*. We also created a PDA scale that assessed PDA enjoyment with four items. We used an imagination task that had the following instructions:

"Imagine walking around [campus/the city center] with your partner. How comfortable do you feel in the following situations?" The items (e.g., We openly show that we are a couple, hold hands, hug, and kiss) were rated on a scale from 1 (*not at all comfortable*) to 7 (*very comfortable*).

**Procedure.** We invited participants via email and an advertisement on campus to fill out an online questionnaire on SoSci Survey [34]. After giving their informed consent, all participants answered demographic and identity-related questions (e.g., sexual orientation). Then, each participant was asked to think about their partner when answering questions about their behavior and feelings in public. Participants were asked to imagine spending time with their (hypothetical) partner and to report how much they enjoyed PDA when walking with this partner in two different contexts (campus vs. city). Participants indicated the gender of their (hypothetical) partner (female/male/other).

Then, each participant reported LGBTQ*-specific minority stressors in the same contexts (campus vs. city). The order in which questions about different contexts were presented was counterbalanced in all instances. We had several more exploratory items and scales. A full account of all the exploratory scales we used can be found in S1 Materials. Moreover, all the original items, scales, and materials we used can be found in the materials on the OSF.

Participants received no monetary reward for their participation. However, psychology students were able to receive course credit. On average, it took participants 14 min ($M = 13.61$, $SD = 3.17$) to complete the survey.

**Sensitivity analysis.** All sensitivity analyses were conducted with G*Power [35]. We found a correlation between the city and campus context of $r = .60$ for minority stress and $r = .80$ for PDA. We calculated the power for the respective effect in a mixed ANOVA with 80% power. In the sample of $N = 78$ participants, we found a context effect on minority stress of $f = 0.14$ (Hypothesis 1, main effect of context), an effect of sexual orientation on PDA of $f = 0.30$ (Hypothesis 2, main effect of sexual orientation), and an effect of $f = 0.10$ for the interaction between context and sexual orientation (Hypothesis 3, interaction effect).

## Results

The 2 (Context: Campus vs. City) x 2 (Sexual and gender identity: LGBTQ* vs. Hetero/Cis) ANOVA with perception of LGBTQ* minority stress as a dependent variable (Fig 1) showed the predicted main effect of context on minority stress, $F(1, 76) = 33.16$, $p < .001$, $\eta_p^2 = 0.30$. The city center was higher in perceived LGBTQ* minority stress ($M = 3.75$, $SD = 1.40$) than the corresponding university campus ($M = 2.90$, $SD = 1.44$), which supported Hypothesis 1. There was also a significant main effect of sexual and gender identity, $F(1, 76) = 7.18$, $p = .009$, $\eta_p^2 = 0.09$, in which Hetero/Cis participants' ratings of LGBTQ* minority stress were lower ($M = 3.00$, $SD = 1.15$) than the ratings of LGBTQ* participants ($M = 3.75$, $SD = 1.30$). This main effect was not predicted and was not the focal interest of our study. However, it seems that participants who are not the target of minority stress tend to underestimate the frequency and likelihood of occurrences. There was no significant interaction between context and sexual and gender identity, $F(1, 76) = 1.76$, $p = .189$, $\eta_p^2 = .023$.

The 2 (Context: Campus vs. City) x 2 (Sexual and gender identity: LGBTQ* vs. Hetero/Cis) ANOVA with PDA enjoyment as the dependent variable yielded significant main effects of context, $F(1, 76) = 7.85$, $p = .006$, $\eta_p^2 = .094$, and sexual and gender identity on PDA, $F(1, 76) =$

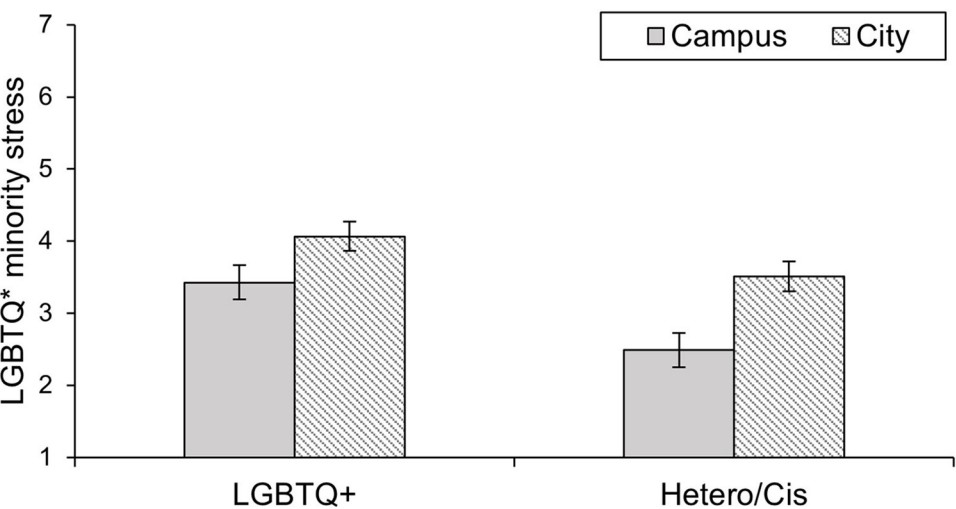

**Fig 1. Perception of LGBTQ* minority stress by LGBTQ* and Hetero/Cis participants in the contexts of a campus and a city.** Error bars represent +/− 1 *SEM*.

4.70, $p$ = .033, $\eta_p^2$ = .058. Overall, PDA enjoyment scores were higher on campus ($M$ = 5.23, $SD$ = 1.48) than in the city ($M$ = 4.98, $SD$ = 1.66). LGBTQ* individuals had lower scores ($M$ = 4.70, $SD$ = 1.49) than Hetero/Cis individuals ($M$ = 5.42, $SD$ = 1.43), supporting Hypothesis 2.

The analysis also yielded a significant interaction effect between context and sexual and gender identity, $F(1, 76)$ = 14.55, $p < .001$, $\eta_p^2$ = .161. The PDA enjoyment of LGBTQ* participants was significantly higher on campus ($M$ = 5.05, $SD$ = 1.49) than in the city ($M$ = 4.35, $SD$ = 1.71), $F(1, 76)$ = 19.40, $p < .001$, $d$ = 0.50, whereas Hetero/Cis participants' PDA enjoyment did not differ significantly between contexts, $F(1, 76)$ = 0.59, $p$ = .446, $d$ = -0.09. Also, the group difference between LGBTQ* and Hetero/Cis individuals disappeared on campus, $F(1, 76)$ = 0.85, $p$ = .360, $d$ = -0.10), whereas it remained significant in the city, $F(1, 76)$ = 9.90, $p$ = .002, $d$ = -0.37. These findings supported Hypothesis 3. Results are displayed in Fig 2. All

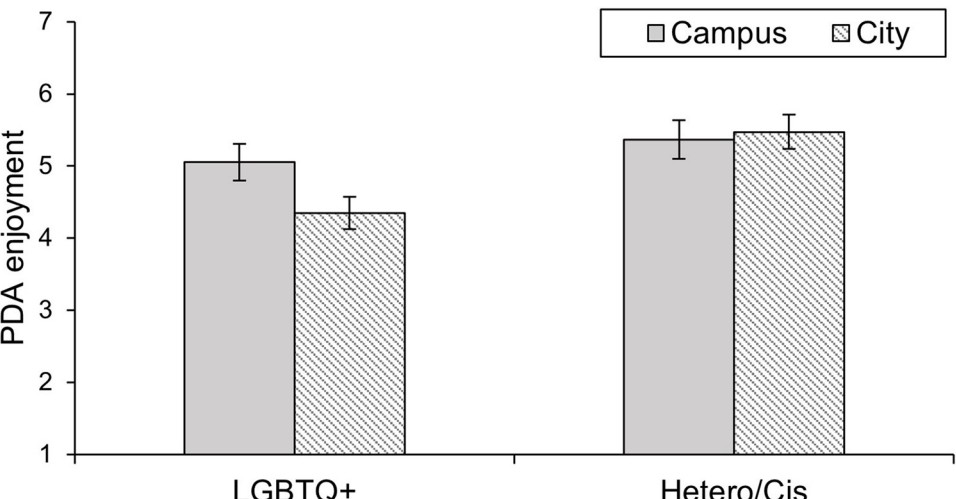

**Fig 2. Reported PDA enjoyment by LGBTQ* and Hetero/Cis participants in the contexts of a campus and a city.** Error bars represent +/− 1 *SEM*.

analyses (in all studies) were conducted with the full sample. Exploratory analyses in which values > |3.5| standard deviations from the group means of LGBTQ* and Hetero/Cis participants were excluded showed that the pattern of results was not affected in any of the studies.

## Discussion

Study 1 showed that minority stress differs between public contexts and that, compared with a university campus, a city center is a context with more minority stress for LGBTQ* individuals (Hypothesis 1). Thus, a city center seems to represent a more harmful social context for LGBTQ* individuals. This first study also indicated that for sexual and gender minority individuals, couple behavior seems to be sensitive to context. LGBTQ* individuals reported enjoying PDA with their (hypothetical) partner substantially less in the city center than on their university campus (Hypothesis 3). However, these results emerged with a relatively small sample size and for a specific university campus and a specific city center. We addressed these shortcomings in Study 2.

## Study 2

Study 2 was a replication of Study 1 aimed at obtaining a larger sample size. It was conducted in Saarbrücken, Germany in 2018. All materials were translated into German.

## Ethics statement

Study 2 and 3 are regarded as minimal risk studies at Saarland University, so that they did not necessitate formal ethical approval. Researchers are expected to conduct their research in line with ethical guidelines provided by the German psychological association. Participants were not fully informed about the goals of the respective study beforehand, because this would have undermined the effects of the experimental manipulations. All participants were thoroughly debriefed via email and received individual feedback on their sexuality profiles upon request.

## Method

**Design.**   Study 2 used the same 2 (Sexual and gender identity: LGBTQ* vs. Hetero/Cis) x 2 (Context: Campus and City) mixed design as Study 1. The dependent variables were, again, minority stress and PDA enjoyment for both contexts.

**Participants.**   The survey was completed by $N$ = 171 university students at a German university with $N_{LGBTQ*}$ = 74 and $N_{Hetero/Cis}$ = 97. Participants were between 17 and 43 years old ($M_{age}$ = 22.43, $SD_{age}$ = 4.11). In this sample, $n$ = 117 participants identified as female (68%), and $n$ = 2 participants identified as other/diverse (1%).

**Independent variables.**   *Sexual and gender identity*. We applied the same rules for separating our sample into LGBTQ* and Hetero/Cis participants as in Study 1.

*Context*. The independent variable context was also the same as in Study 1. We differentiated between the local university campus and the corresponding city center and defined both contexts as easily accessible public areas. The imagination task and the instructions remained the same as well.

**Dependent variables.**   Measures for Study 2 included the same scales for LGBTQ* minority stress (SMSS) and for PDA enjoyment as in Study 1. Reliabilities can be found in Table 3.

**Procedure.**   We invited participants via a campus advertisement and university-related Facebook groups to fill out an online questionnaire on SoSciSurvey. The procedure resembled the one in Study 1. After giving informed consent and answering demographic and identity-

**Table 3. Summary of correlations, reliabilities, means, and standard deviations for scores on scales for PDA and minority stress in Study 2.**

| Measure | 1 | 2 |
|---|---|---|
| 1. PDA Enjoyment Scale | .93 | — |
| 2. SMSS | -.003 | .87 |
| | (.973) | |
| M | 6.02 | 3.02 |
| SD | 1.09 | 1.10 |

*Note.* SMSS = Short Minority Stress Scale. Reliabilities (Cronbach's alpha) are presented on the diagonal, *p*-value is presented in parentheses.

related questions (e.g., sexual orientation), participants, again, were asked to imagine spending time with a (hypothetical) partner and to report how much they enjoyed PDA in the same two contexts that we used in Study 1. Afterwards, each participant answered questions about minority stress. The order of the context-related questions was counterbalanced. A full account of all the exploratory scales we used can be found in S2 Materials. Moreover, all the original items, scales, and materials that we used can be found in the materials on the OSF. Participants were given no monetary reward for their participation. However, psychology students were able to receive course credit. On average, it took participants 14 min (*M* = 13.56, *SD* = 3.67) to complete the survey.

**Sensitivity analysis.** We found a correlation between the city and campus context of *r* = .61 for minority stress and *r* = .81 for PDA. We calculated the power for the respective effect in a mixed ANOVA with 80% power. In the sample of *N* = 171 participants, we found an effect of *f* = 0.10 for minority stress (Hypothesis 1, main effect of context), an effect of *f* = 0.20 for PDA (Hypothesis 2, main effect of sexual orientation), and an effect of *f* = 0.07 for the interaction between context and sexual orientation (Hypothesis 3, interaction effect).

## Results

The 2 (Context: Campus vs. City) x 2 (Sexual and gender identity: LGBTQ* vs. Hetero/Cis) ANOVA with perception of LGBTQ* minority stress as the dependent variable showed the predicted main effect of context on minority stress, $F(1, 169) = 185.76$, $p < .001$, $\eta_p^2 = 0.52$. The city center was higher in perceived LGBTQ* minority stress (*M* = 3.59, *SE* = 0.10) than the corresponding university campus (*M* = 2.42, *SE* = 0.09), which, again, supported Hypothesis 1. The main effect of sexual and gender identity, $F(1, 169) = 0.25$, $p = .621$, $\eta_p^2 = .001$, and the interaction between context and sexual and gender identity, $F(1, 169) = 1.55$, $p = .214$, $\eta_p^2 = .009$, were not statistically significant. The results are displayed in Fig 3.

The 2 (Context: Campus vs. City) x 2 (Sexual and gender identity: LGBTQ* vs. Hetero/Cis) ANOVA with PDA enjoyment as the dependent variable yielded a significant main effect of sexual and gender identity, $F(1, 169) = 4.67$, $p = .032$, $\eta_p^2 = .027$ (confirming Hypothesis 2). LGBTQ* individuals had lower scores (*M* = 5.73, *SD* = 1.20) than Hetero/Cis individuals (*M* = 6.13, *SD* = 1.19). However, the main effect of context was not significant, $F(1, 169) = 1.11$, $p = .293$, $\eta_p^2 = .007$. The interaction effect, which we predicted in Hypothesis 3, was not significant either, $F(1, 169) = 0.59$, $p = .443$, $\eta_p^2 = .003$. Accordingly, the comparison between the campus and the city for the LGBTQ* subsample remained nonsignificant as well, $F(1, 76) = 0.04$, $p = .851$, $d = -0.01$. The results are shown in Fig 4.

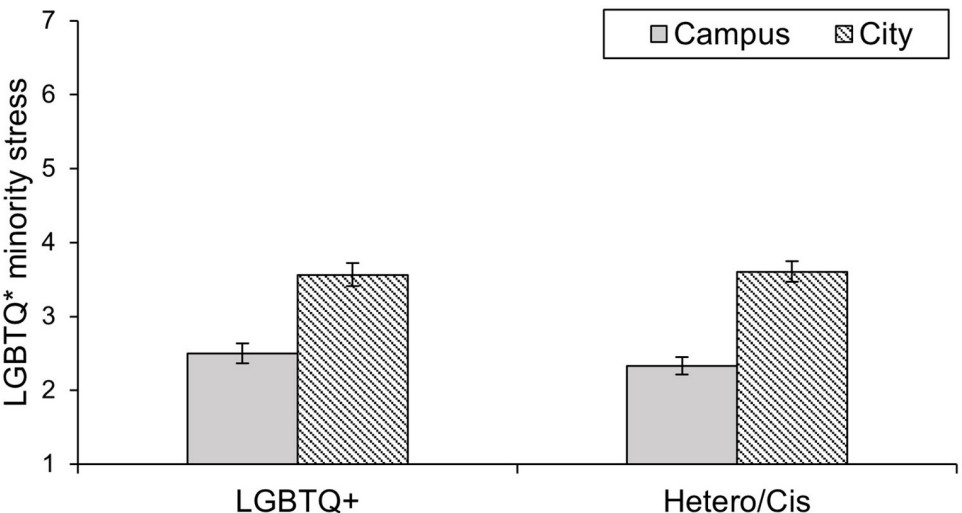

**Fig 3. Perception of LGBTQ* minority stress by LGBTQ* and Hetero/Cis participants in the contexts of a campus and a city in Study 2.** Error bars represent +/− 1 *SEM*.

## Discussion

The findings of Study 2 further supported the notion that the city center and campus differ with respect to LGBTQ* minority stress. The results showed that, again, the city was associated with substantially more minority stress than the campus (Hypothesis 1).

The finding that LGBTQ* individuals show less PDA than heteronormative individuals was replicated as well (Hypothesis 2). However, LGBTQ* participants' PDA enjoyment did not differ between the two contexts. Thus, we could not replicate the finding that LGBTQ* participants enjoy PDA less in the city versus on campus (Hypothesis 3). Interestingly, this could mean that LGBTQ* participants feel more stigmatized and more likely to experience discrimination or harassment in the city, but this does not influence the way they enjoy sharing

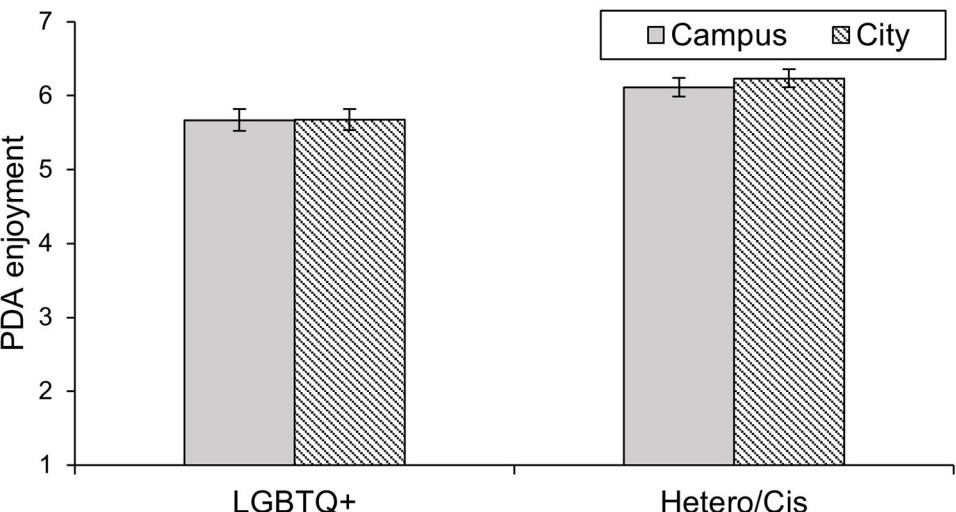

**Fig 4. Reported enjoyment of PDA by LGBTQ* and Hetero/Cis participants in the contexts of a campus and a city in Study 2.** Error bars represent +/− 1 *SEM*.

affection with their partner. Another way to interpret this finding could be that, instead, more PDA (and even more PDA enjoyment) leads to more concern for safety and more awareness that each act of PDA has the potential to attract unwanted attention. As such, Muraco [36] reported that daily PDA was positively associated with concurrent and lagged relationship satisfaction but also with concurrent and lagged (fear of) harassment and unsafe situations. Another explanation could be that the direct effect of context on PDA was suppressed by the indirect effect of minority stress. If the city context is generally associated with more PDA (as suggested by the results of the Hetero/Cis group), then the negative impact of minority stress could even things out, and a null effect, as found here, may occur. Accounting for minority stress may therefore yield the expected indirect effect (campus > city) as well as the unexpected direct effect (city > campus) for LGBTQ* participants. We did not test this proposition in this study because the sample size was too small for a mediation analysis. However, this led to a planned within-mediation test in Study 3.

A limitation of this study, one which could also account for the fact that we were not able to replicate the findings from Study 1, is the differentiation between LGBTQ* and Hetero/Cis participants. The Study 2 sample included a large number of participants who were not exclusively attracted to one gender (e.g., bisexual). For the PDA assessment, we had asked participants to identify the gender of their real or imagined partner. Some of these LGBTQ* participants identified an different-sex partner when they answered these questions ($n_{\text{LGBTQ*|different-sex}}$ = 27). However, phenotypically, such couples are in a heteronormative relationship. We would expect LGBTQ* participants who reported being with a different-sex partner to enjoy PDA more than participants who were phenotypically with a same-sex partner. Thus, it is possible that we could not find an effect of sexual orientation on PDA enjoyment because of the classification rules we had used. Therefore, in Study 3, we looked at the phenotype of the relationship for which participants reported PDA.

## Study 3

Study 3 ($N_{total}$ = 268, $n_{same-sex}$ = 80) was conducted in Saarbrücken, Germany in 2019. In contrast to Studies 1 and 2, participants were recruited not only at the respective university and city but nationwide. A distribution of participants' locations can be found in S1 Table. We made several changes in Study 3. First, because the results of Studies 1 and 2 were inconsistent in some respects, we aimed to refine how we defined the different groups to clarify the effects we found. We reasoned that a person's sexual orientation (e.g., gay) and a person's relationship type (e.g., same-sex) should have similar effects on minority stress. However, whether couples experience minority stress (or the fearful anticipation thereof) should strongly depend on whether they are a same-sex or a different-sex couple. In other words, a person with a same-sex partner may be more likely to be afraid that they will be attacked for exhibiting PDA than a person with a different-sex partner. This supposition is also in line with the assumption that there is a difference between something that happens to the couple and something that happens to each partner as an individual [4, 37]. Therefore, we used the phenotype of the relationship (i.e., same-sex vs. different-sex partners) to classify the groups. Second, we included a neutral control context (i.e., private at home) to ensure that same-sex and different-sex participants did not inherently differ in their relationship behavior, or more specifically, in the extent to which they display affection. We used a private context because it is assumed to be freer from direct social influences, from the presence of other people, and from external minority stress. Hence, we had no reason to assume that same-sex and different-sex couples would differ significantly from one another in this context. Third, we used peer-reviewed PDA and minority stress scales that included more items. We also differentiated between different components

of minority stress, namely, minority stress, perceived danger, marginalization, and PDA (i.e., PDA enjoyment and PDA frequency). Finally, we used a combination of quantitative and qualitative methods to further investigate the context effects that emerged in Studies 1 and 2. This mixed-methods approach is recommended for contextualized phenomena [38] and for the diverse field of minority research [39], thus making it a reasonable choice for our topic. We used an open-ended question and thematic analysis to investigate participants' thoughts on why their personal PDA differed between the two contexts.

Besides testing Hypotheses 1, 2, and 3 again, we additionally made assumptions about the newly added private context. We expected that PDA would be lower than private displays of affection for all individuals (Hypothesis 4a) and that, in a private context, there would be no difference between same-sex and different-sex couples concerning their displays of affection (Hypothesis 4b).

If we ended up finding support for Hypothesis 3 (i.e., lower PDA in the city than on campus for same-sex couples), we would further assume that minority stress is the process behind this effect. We hypothesized that minority stress would mediate the effect of public contexts on PDA for same-sex couples (Hypothesis 5, mediation effect).

## Method

**Design.** We used a 3 (Context: Campus vs. City vs. Private) x 2 (Relationship type: Same-Sex vs. Different-Sex) mixed design.

**Participants.** The survey was completed by $N$ = 268 university students at more than 30 different German universities with $n_{same-sex}$ = 80, $n_{different-sex}$ = 188, $n_{LGBTQ*}$ = 139, and $n_{Hetero/Cis}$ = 129 participants. Participants were between 18 and 35 years old ($M_{age}$ = 23.38, $SD_{age}$ = 3.78), and $n$ = 141 (53%) participants reported being in a committed relationship. In this sample, $n$ = 168 participants identified as female (62%), $n$ = 25 participants identified as other/diverse (9%), and $n$ = 75 participants identified as male (28%).

**Independent variables.** *Relationship type*. Because our research focused on distal stressors, which are more pronounced when people are with a same-sex vs. different-sex partner, we separated participants into two groups on the basis of whether they answered our questions with a same-sex or a different-sex partner in mind. As Meyer [4] pointed out: If someone is perceived as gay or lesbian, they will be more likely to be confronted with minority stressors regardless of whether this perception is aligned with their actual sexual and gender identity (e.g., bisexual). As the terminology of LGBTQ* includes bisexuals with no regard to their partner's gender, we henceforth changed the names of our groups from "sexual and gender identity" (LGBTQ* vs. Hetero/Cis) to "relationship type" (same-sex vs. different-sex). For example, a bisexual female person with a male partner (both cisgender) would most likely be in the minority group in Studies 1 and 2 because they self-identified as LGBTQ* but might not be in the minority group in Study 3 if they indicated they had a different-sex partner. However, we do point out that bisexual and pansexual people are, of course, a minority group and are frequently confronted with identity denial and erasure [40]. We are also aware of the fact that it might have been more difficult for bisexual and pansexual participants to classify their imagined partner.

*Context*. As in Studies 1 and 2, we differentiated between the campus and city as two public places. In Study 3, we added a third context—a private setting. The instructions were: "Imagine you are with your partner in a private place. A private place is defined as a context in which no one other than you and your partner is or will be present (e.g., at home)."

**Dependent variables.** Table 4 presents the means, standard deviations, and internal consistency scores (Cronbach's alpha) of the dependent variables.

**Table 4. Summary of correlations, reliabilities, means, and standard deviations for the physical affection and minority stress scales in Study 3.**

| Measure | 1 | 2 | 3 | 4 | 5 |
|---|---|---|---|---|---|
| 1. Frequency of physical affection (APPPA) | .87 | — | — | — | — |
| 2. Enjoyment of physical affection (APPPA) | .74** (< .001) | .94 | — | — | — |
| 3. SMSS | -.10 (.387) | -.10 (.365) | .87 | — | — |
| 4. Perceived Danger | -.28* (.013) | -.37** (.001) | -.50** (< .001) | .84 | — |
| 5. Marginalization | -.30** (.008) | -.36** (.001) | -.38** (.001) | -.53** (< .001) | .65 |
| *M* | 3.84 | 5.01 | 2.97 | 3.19 | 2.76 |
| *SD* | 1.25 | 1.36 | 0.98 | 0.77 | 0.90 |

*Note.* APPPA = Assessment of Public and Private Physical Affection; SMSS = Short Minority Stress Scale. Reliabilities (Cronbach's alpha) are presented on the diagonal, *p*-values are presented in parentheses.

* *p* < .05.

** *p* < .01.

*Minority stress*. In addition to the SMSS, which we used in Studies 1 and 2, we included two peer-reviewed measures of minority stress. The perceived danger subscale was taken from Brady's adapted version of the fear of heterosexism scale [41]. It consists of five items measured on a 7-point Likert scale. An example is "I feel vulnerable to violence from strangers." Marginalization was assessed with two items that were taken and adapted from Lehmiller and Agnew's marginalization scale [5], which originally contained four items. We changed the items so it would be possible to answer the questions about societal (dis-) approval for each public context. The items were: "My relationship is generally accepted and/or approved of [on campus vs. in the city center]" and "I believe that most other people [on campus vs. in the city center] (whom I do not know) would generally disapprove of my relationship." Because findings on differences in perception between LGBTQ* and Hetero/Cis participants concerning LGBTQ* minority stress could not be replicated in Study 2 and the fact that this was not the focal interest of our research, the minority stress items were only answered by same-sex participants. Due to a technical error, the first $n_{\text{same-sex}} = 4$ participants were not asked the questions about minority stress, but the error was fixed after these four. Hence, the sample for analyses concerning minority stress consisted of $n_{\text{same-sex}} = 76$ participants.

*Physical affection*. Concerning physical affection (which we still call PDA in public settings but refer to as physical affection now because we added a private setting in this study), in addition to the enjoyment of physical affection in Study 3, we also assessed the frequency of physical affection. We did so to further differentiate between different components of physical affection.

We used the Assessment of Public and Private Physical Affection [3] to assess the enjoyment and frequency of physical affection, which included seven physical affection items (i.e., holding hands, kissing on the lips, kissing on the face, cuddling/holding, sitting on each other's laps, and hugging/embracing). As we asked for both the frequency and enjoyment of physical affection, this resulted in 14 items per page and context.

**Qualitative measures.** We aimed to investigate participants' thoughts about why their personal PDA differed between the two public contexts. To do so, we administered an open-ended question that asked participants about their reasons for preferring one public context over the other. The survey program compared the individual's PDA score on campus with their PDA score in the city. On the basis of this comparison, participants received a note stating that their answers indicated that they felt more comfortable and showed more PDA in Context A (e.g., campus) than in Context B (e.g., city). A subsequent open-ended question

investigated the individual's reasoning for why this could be the case. The exact wording was: "Your answers indicate that you feel more comfortable about PDA in the city than on campus (or vice versa) and that you tend to show more affection in the city context. Do you agree that this is true? Why is this the case for you personally? Please give us a brief explanation."

**Procedure.** Participants were invited via email, an online advertisement on Facebook or Reddit, or via advertisements on two local campuses or in two city centers to take part in an online study. In contrast to Studies 1 and 2, we targeted students attending any (German) university. After providing informed consent, all participants answered questions about their demographics and (sexual and gender) identity. After answering questions about their identity, participants were again asked to imagine spending time with their (hypothetical) partner and to report how much they enjoyed PDA when walking with this partner in two different contexts (campus vs. city) and also, how often they showed this behavior (e.g., kissing, holding hands) in public. Each participant was asked to explicitly state whether their (hypothetical) partner was of the same or a different sex. Also, only participants with a same-sex (hypothetical) partner were given questions about LGBTQ* minority stress afterwards. These questions were again asked for two contexts (campus vs. city). Questions regarding different contexts were presented in a counterbalanced order. A subsequent open-ended question explored participants' reasons for context differences. A full account of all the exploratory scales we used can be found in S3 Materials. Moreover, all the original items, scales, and materials we used can be found in the materials on the OSF.

No monetary reward was offered for participation. Psychology students received course credit for their participation. All participants were able to enter a lottery to win one of two books on relationship health. As further compensation, we offered personalized feedback on their sexual orientation profile. It took participants an average of 16 min ($M = 15.90$, $SD = 6.11$) to complete the survey.

**Sensitivity analysis.** We conducted sensitivity analyses for Study 3 for repeated-measures MANOVAs with 80% power because each dependent variable had more than one subscale.

The correlations between the campus and the city contexts for the three minority stress subscales were $r = .49$ for perceived danger, $r = .40$ for marginalization, and $r = .57$ for SMSS. Using the lowest and most conservative correlation of $r = .40$, in the sample of $n_{same-sex} = 76$ participants, we found a context effect of $f = 0.18$ on minority stress (Hypothesis 1, main effect).

For the enjoyment and frequency of physical affection in three contexts, the lowest correlation was $r = .47$ for the frequency of displays of physical affection in private and on campus, whereas the highest correlation was $r = .80$ for the enjoyment of displays of physical affection in the city and on campus. For the lowest and most conservative correlation, in the sample of $N = 268$ participants, we found an effect of $f = 0.13$ with 80% power for the enjoyment and frequency of physical affection (Hypothesis 2, main effect) and an interaction effect between relationship type and context of $f = 0.18$ with 80% power (Hypothesis 3).

## Results

**Quantitative results.** The 2 (Context: Campus vs. City) x 1 (Relationship type: same-sex) MANOVA with the three minority stress scales (i.e., perceived danger, marginalization, and SMSS) as the dependent variables yielded a significant main effect of context in the multivariate analysis, $F(3, 72) = 76.21$, $p < .001$, $\eta_p^2 = 0.96$. Supporting Hypothesis 1, the campus was rated as a place with substantially less minority stress (Fig 5). Univariate analyses revealed that the perception of LGBTQ* minority stress was significantly lower on campus than in the city center on all scales, namely, perceived danger, $F(1, 74) = 121.01$, $p < .001$, $\eta_p^2 = 0.62$,

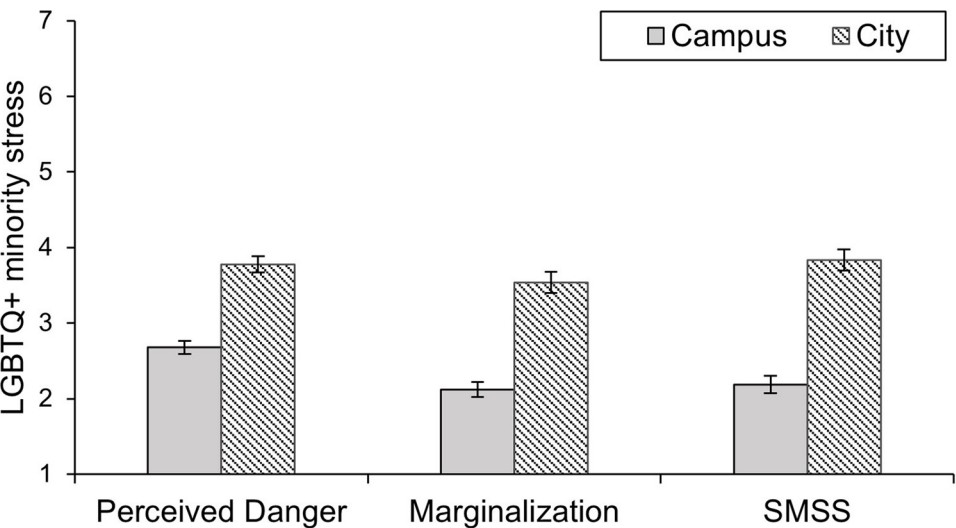

**Fig 5. Reported LGBTQ\* minority stress for the contexts of a campus and a city in Study 3, with perceived danger, marginalization, and the SMSS as different outcome measures.** Error bars represent +/− 1 *SEM*.

marginalization $F(1, 74) = 102.18$, $p < .001$, $\eta_p^2 = 0.58$, and the SMSS, $F(1, 74) = 185.19$, $p < .001$, $\eta_p^2 = 0.71$. This finding demonstrates that this effect is not limited to one minority stress scale.

The 3 (Context: Campus vs. City vs. Private) x 2 (Relationship type: Same-Sex vs. Different-Sex) MANOVA with the enjoyment and frequency of physical affection as the dependent variables also yielded a significant main effect of context in the multivariate analysis, $F(4, 1060) = 104.37$, $p < .001$, $\eta_p^2 = 0.28$, using Pillai's trace. There was also a significant effect of relationship type, $F(2, 264) = 8.01$, $p < .001$, $\eta_p^2 = 0.06$. Supporting Hypothesis 2, individuals with a same-sex partner generally had lower enjoyment ($M = 5.13$, $SD = 0.12$) and frequency ($M = 4.32$, $SD = 0.12$) of physical affection than those with a different-sex partner ($M = 5.58$, $SD = 0.08$ for enjoyment; $M = 4.41$, $SD = 0.08$ for frequency). In univariate tests, however, the main effect of relationship type was only significant for enjoyment, $F(1, 265) = 9.41$, $p = .002$, $\eta_p^2 = 0.03$, but not for frequency, $F(1, 265) = 1.46$, $p = .519$, $\eta_p^2 < 0.01$.

The MANOVA also showed a significant interaction between relationship type and context, $F(4, 1060) = 6.29$, $p < .001$, $\eta_p^2 = 0.02$, using Pillai's trace. Univariate tests showed significant interaction effects for enjoyment, $F(1.79, 264) = 11.45$, $p < .001$, $\eta_p^2 = 0.04$, and frequency, $F(1.78, 264) = 4.45$, $p < .001$, $\eta_p^2 = 0.02$. We thus compared the campus versus the city for same-sex participants. This paired comparison was significant for enjoyment, $F(4, 262) = 54.35$, $p = .015$, $d = 0.13$, but not for frequency, $F(4, 262) = 54.35$, $p = .281$, $d = 0.06$. The direction of the effect was as predicted: Same-sex participants reported more enjoyment of PDA on campus ($M = 4.73$, $SD = 0.16$) than in the city ($M = 4.49$, $SD = 0.16$), but PDA was not significantly more frequent on campus than in the city.

Concerning participants with a different-sex partner, our analyses revealed significant paired comparisons for the campus versus the city context for PDA enjoyment and frequency, $F(4, 262) = 54.35$, all $ps < .001$, all $ds < -0.23$. For individuals in a heteronormative relationship, the city rather than the campus was the context associated with more enjoyable and frequent PDA.

Thus, Hypothesis 3 was supported for PDA enjoyment but not for PDA frequency for same-sex relationships, whereas context-sensitivity emerged in the opposite direction for enjoyment and frequency for different-sex relationships. Results are displayed in Fig 6.

Further analyses showed that there was no difference between same-sex and different-sex relationships in the campus context both for PDA enjoyment, $F(2, 264) = 4.01$, $p = .110$ and frequency, $F(2, 264) = 4.01$, $p = .580$. However, the difference between same-sex and different-sex relationships was significant in the city for PDA enjoyment, $F(2, 264) = 12.64$, $p < .001$, $d = -0.31$ and frequency $F(2, 264) = 12.64$, $p = .050$, $d = 0.13$.

Concerning the newly added private context, univariate tests showed that the context effect was significant for both the enjoyment, $F(1.79, 264) = 236.84$, $p < .001$, $\eta_p^2 = 0.47$, and frequency, $F(1.78, 264) = 306.28$, $p < .001$, $\eta_p^2 = 0.54$, of physical affection. For all participants, PDA was significantly less enjoyable, $M = 5.01$, $SD = 1.36$, $F(1, 266) = 354.06$, $p < .001$, $\eta_p^2 = 0.57$, and significantly less frequent, $M = 3.84$, $SD = 1.25$, $F(1, 266) = 592.87$, $p < .001$, $\eta_p^2 = 0.64$, compared with private displays of affection ($M = 6.32$, $SD = 0.93$ for enjoyment; $M = 5.45$, $SD = 1.09$ for frequency), which confirmed Hypothesis 4a.

Using Pillai's trace, there was no significant effect of type of relationship on displays of affection for the private context on multivariate tests, $F(2, 265) = 2.23$, $p = .109$, $\eta_p^2 = 0.02$. Univariate tests showed that for both the enjoyment, $F(2, 266) = 2.61$, $p = .108$, $\eta_p^2 = 0.01$, and frequency, $F(2, 266) = 0.06$, $p = .813$, $\eta_p^2 < 0.01$, of displays of affection, there was no significant difference between participants in a same-sex versus a different-sex relationship in a private context. Taken together, this finding supported Hypothesis 4b and the assumption in all three studies that, given a stress-free environment, same-sex relationships do not differ from heteronormative ones in terms of PDA.

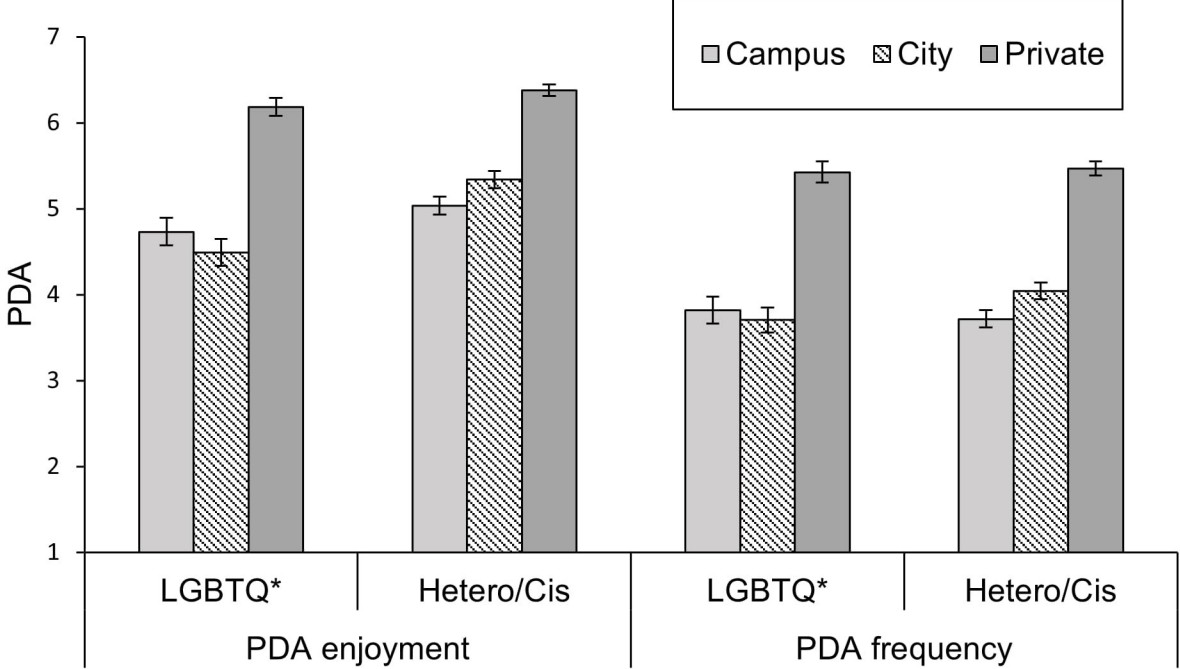

**Fig 6. Reported enjoyment and frequency of physical affection by same-sex and different-sex participants in the contexts of a campus, a city, and a private setting in Study 3.** Error bars represent +/− 1 *SEM*.

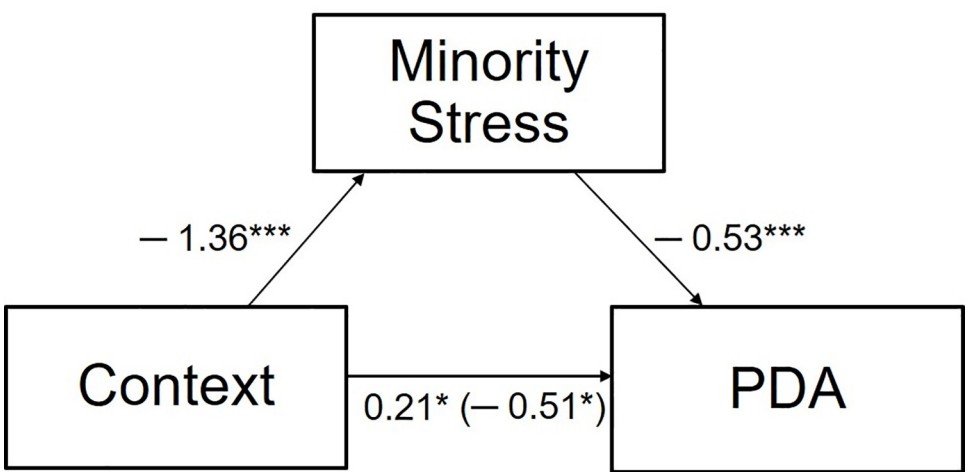

**Fig 7. A within-participants mediation analysis in which minority stress mediates the effect of context on PDA.**
Context is coded 0 = campus and 1 = city. * $p < .05$. *** $p < .001$.

A two-condition within-participant mediation analysis using MEMORE V2.1 [42] (5,000 bootstrapping confidence intervals) revealed that the effect of a public context (campus vs. city) on PDA enjoyment was mediated by perceived minority stress (supporting Hypothesis 5). The analysis included answers from $n_{same-sex} = 76$ participants who had previously indicated a same-sex relationship and answered questions about minority stress. Results are displayed in Fig 7.

For this analysis, PDA was calculated as the mean of PDA enjoyment and PDA frequency, and minority stress was calculated as the mean of the SMSS and the perceived danger and marginalization scales. There was a significant indirect effect of public context on PDA, $b = 0.72$, $SE = 0.19$, BCa CI [0.36, 1.13]. There was a significant total effect of context on PDA, $b = 0.21$, $SE = 0.10$, $t(1, 75) = 2.03$, $p = .046$, BCa CI [0.004, 0.42]. Interestingly, there was also a significant direct effect of context on PDA, $b = -0.51$, $SE = 0.20$, $t(1, 73) = -2.55$, $p = .013$, BCa CI [-0.93, -0.14]. These findings indicate that the campus context is associated with more (enjoyable) PDA for same-sex couples. However, it must be noted that this effect went in the opposite direction of the total effect. Whereas we would have expected the significant difference between public contexts for PDA to disappear in the direct effect, this effect reversed direction when we examined it without the influence of minority stress. This indicates that, given a minority stress-free environment, same-sex couples would feel more comfortable and show more PDA in the city, the context that is otherwise associated with less PDA. This notion ties in with the finding that different-sex couples, who are unlikely to be affected by distal LGBTQ* minority stress, feel significantly more comfortable and show more PDA in the city compared with on campus.

**Qualitative results.** The open-ended question concerning PDA context preference yielded 7,456 words. Most participants (54.5%, $n_{city} = 146$) preferred the city context (as indicated by the sum scores for PDA enjoyment and frequency), whereas 39.5% ($n_{campus} = 106$) preferred the campus context. The remaining 6% ($n_{equal} = 16$) had equal scores in the two public contexts. To further analyze the contents of the answers, we followed the six phases for thematic analyses as defined by Braun and Clarke [43] as displayed in Fig 8.

In Phase 1, we found four preliminary topic patterns that seemed to be dominating the answers. The first topic pattern was "liberality of context," which seemed to describe one public context as more open-minded or safe and the other one as more judgmental or potentially dangerous. These answers included mentions of homophobia, fear of homophobic incidents,

| Phase 1 | Familiarizing yourself with your data |
| Phase 2 | Generating initial codes |
| Phase 3 | Searching for themes |
| Phase 4 | Reviewing themes |
| Phase 5 | Defining and naming themes |
| Phase 6 | Producing the report |

**Fig 8. A step-by-step guide to thematic analysis as defined by Braun and Clarke [43].**

and feeling more protected in one of the two public contexts. The second topic pattern was "professionality and anonymity." This pattern was represented by answers given by respondents who wanted to behave in a "work-appropriate" manner in one context or who could show PDA more openly in a context in which they spent personal/leisure time and felt more anonymous. The third topic pattern was "familiarity." Answers included feeling more comfortable showing PDA in a more familiar public context, also (but not exclusively) with respect to people the participants knew and trusted. The fourth topic pattern included answers from participants who clearly stated that, really, they did not see a difference between the two public contexts.

Phase 2 included coding the data set, which was aimed at sorting the data into meaningful groups (i.e., codes). For the coding process, we used the program MAXQDA 2020 [33] (VERBI Software, 2019). The original file with the coded text data can be found on the OSF. Because the codes tended to be more differentiated than the themes, they generated a much longer list. In this initial coding process, 41 codes were created and assigned to the (chunks of) open-ended answers. Some codes seemed like they might form an overarching theme later and were therefore coded with the same color (Phase 3). An example is the code "fear of rejection," whose contents seemed to be closely related to the codes "absence of rejection," "fear of negative evaluation," "disparaging remarks," and "disparaging looks." These codes might also be aligned with "potential danger," "absence of danger," "feeling safe/protected," "safe space," "open-minded atmosphere," "tolerance and acceptance," "absence of tolerance and acceptance," "discrimination," "absence of discrimination," and "(visibility of) diversity."

In Phases 4 and 5, preliminary topic patterns from Phase 1 were revised by using the codes from Phase 2. This process yielded five themes, as all codes could be assigned to either the *protective nature of the context*, the *harmful nature of the context*, the *professional nature of the context*, the *private nature of the context*, or the *(general) public nature of the context*. Table 5 shows these five themes with their corresponding contents.

Compared with the initial pattern from Phase 1, these themes were able to distinguish between the protective nature of one context and the harmful nature of another, both of which could be employed as reasons to favor one context over another. Interestingly, some participants answered the initial question about why their PDA was more frequent and enjoyable in one context by explaining why their PDA was less frequent or enjoyable in the *other* context. The following answer is an example of a participant employing both lines of argumentation:

> *"Because there are younger and more open-minded people on campus who are more laid-back about same-sex couples or who do not bother about or interfere with other people's lives that much as is the case with older people in the city."*

This example shows that both the positive aspects of one context *and* the negative aspects of the other each contributed to the preference for or the dislike of one context concerning PDA.

**Table 5. Themes on negotiating PDA in different public contexts.**

| Main topics (themes) | Defining contents (codes) |
|---|---|
| Protective nature of the context | Feeling safe/protected |
| | Safe space |
| | Absence of rejection |
| | Absence of danger |
| | Open-minded atmosphere |
| | Tolerance and acceptance |
| | Absence of discrimination |
| | (Visibility of) diversity |
| | Young people |
| | People who think/are like me |
| | Educated people |
| | Liberal/tolerant people |
| Harmful nature of the context | Fear of rejection |
| | Fear of negative evaluation |
| | Disparaging remarks |
| | Disparaging looks |
| | Potential danger |
| | Absence of tolerance & acceptance |
| | Discrimination |
| | Conservative people |
| | People who are older (than me) |
| | Attracting unwanted attention |
| Professional nature of the context | Professionality (wanting to be professional) |
| | Fear of being unprofessional |
| | Absence of togetherness |
| | Being embarrassed in front of people I know |
| | Fear of gossip |
| | Absence of leisure time |
| | Inappropriate in company |
| | Only a few/fewer opportunities |
| | Absence of familiarity |
| Private nature of the context | Anonymity |
| | Togetherness |
| | Leisure time |
| | More private |
| | Friends & familiar people |
| | Familiarity (familiar space) |
| | People are indifferent to PDA |
| | Lots of/more opportunities |
| (General) public nature of the context | Generally uncomfortable in public |
| | No difference ("both are public contexts") |

Because group differences between individuals with same-sex and different-sex partners were important in the current research, we investigated which themes were employed in which groups of answers. To this end, each answer was considered one data item and was color-coded for preferred context (city vs. campus vs. equal scores), for the type of relationship (same-sex vs. different-sex), and for the sexual and gender identity of the respondent (LGBTQ*

vs. Hetero/Cis). Both the *protective nature of the context* and the *harmful nature of the context* were predominant in answers by LGBTQ* participants who indicated a same-sex relationship and favored the campus context. In answers from participants who favored the city context, the *professional nature of the context* and the *private nature of the context* were especially important. Interestingly, these components were the most important among all groups who favored the city context (LGBTQ*, Hetero/Cis, same-sex, and different-sex alike).

## General discussion

Our results show that LGBTQ* minority stress differed substantially between the campus and the city contexts in all three studies (supporting Hypothesis 1). This strongly indicates that minority stress is indeed sensitive to context (Research Question 1). Moreover, a context with increased minority stress seems to impair same-sex couples' enjoyment during displays of affection, meaning they enjoy PDA less in more harmful public contexts (Studies 1 and 3). Meanwhile, such a harmful effect could not be found for the frequency in which same-sex couples show affection (Study 3, partly supporting Hypothesis 3). These findings indicate that the perception of minority stress translates at least partly into couple behavior (Research Question 2). We also showed that PDA is generally less frequent and less enjoyable than private displays of affection (Hypothesis 4a) and that same-sex couples' private displays did not differ from those of different-sex couples (Hypothesis 4b).

Finally, LGBTQ* minority stress accounted for the lower PDA of same-sex couples in the city compared with on campus in a within-participants mediation analysis (supporting Hypothesis 5). This result indicates that a fear of or a confrontation with minority stress in a certain public context could inhibit LGBTQ* couples' behavior.

Our qualitative results show that a university campus might be more LGBTQ* friendly than a corresponding city center, whereas our quantitative results show that a city center seems to be more harmful for LGBTQ* couples. However, given two social contexts that are equally protective in terms of LGBTQ* friendliness, it seems like same-sex couples would even prefer the opposite context (i.e., the city). This ties in with the finding that individuals in different-sex relationships, who are less likely to be affected by LGBTQ* minority stress, prefer the city over the campus in terms of PDA with their partner. Reasons for this could be found in our thematic analysis, which showed that the city was associated with leisure time, a feeling of togetherness with a partner and, most importantly, the feeling of being anonymous. Meanwhile, the campus setting was considered a more professional context in which PDA might be inappropriate.

Our thematic analysis further investigated what the protective value of a social context could be (Research Question 3). We found that an open-minded atmosphere, the absence of danger, and people who are liberal, tolerant, or of the same age were reasons to be more comfortable and open in a social context. This added to our quantitative results, which highlighted the effect of harmful social factors by demonstrating the importance of the protective nature of a context for LGBTQ* individuals and couples.

Taken together, our quantitative and qualitative results concerning contextual preferences for PDA go hand in hand. From their unique perspective, each shows that when equaling out the harmful and protective natures of the two public contexts we investigated (e.g., by taking away the positive impact of protectiveness on campus or adding this protectiveness to the city context), there might be more reasons to openly show and enjoy PDA in a city center rather than on a university campus. This shows that, even in times when restrictions have been eased and attitudes have become more LGBTQ*-friendly in most European countries, the impact of minority stress on same-sex couples remains strong. Creating, promoting, and maintaining

protective safe spaces for LGBTQ* individuals and couples in everyday public areas—beyond LGBTQ*-specific areas such as support centers—might therefore be the next step toward achieving a more inclusive atmosphere that promotes LGBTQ* well-being. To this end, further research is needed. However, our studies give some insight into the fact that it might be the little things that count. As some arguments go, even seeing rainbow flags or advertisements for LGBTQ*-related events on campus can promote a feeling of safety and openness connected to the context. This shows that making diversity (more) visible may be advisable, as it is a small but effective act.

## Limitations

One limitation of our studies is that we used student samples. Although we were able to newly establish the existence of context effects on LGBTQ* minority stress and PDA, participants with a less educated background may respond differently. A second limitation concerns the hypothetical nature of our study. Alternatives are discussed below. Third, we chose to investigate two very specific contexts: a city center and a university campus. Of course, this choice of contexts may seem arbitrary to some degree, and other researchers may replace them with other contexts. But we recognized certain advantages that led us to stick with these two choices. Both contexts are familiar to an average participant in a student sample. More specific settings, such as a (gay) bar or a train station, are contexts only a proportion of individuals regularly move in, which may lead to answers that are primarily based on speculation. Other contexts could be a family gathering versus a gathering with friends, but these contexts come with the disadvantage that their spatial setting often changes, and they are more individual and less affected by standard societal norms. Future research should investigate whether the current findings generalize to other liberal versus nonliberal contexts.

Finally, we used one method to group participants in Studies 1 and 2 and another in Study 3. Whereas Studies 1 and 2 focused on (self-assigned) LGBTQ* participants as the minority group, Study 3 examined participants who self-assigned themselves to being in a same-sex relationship. Because we asked similar questions in all three studies, we were able to conduct an exploratory analysis on the data from Studies 1 and 2 with the classification method used in Study 3 and to analyze the data from Study 3 with the method used in Studies 1 and 2. These analyses showed two things: First, when we employed the classification method from Study 3 in Studies 1 and 2, the size of the sample of participants in a same-sex relationship was too small for us to have confidence in the results ($n_{\text{same-sex}}$ = 17 for Study 1 and $n_{\text{same-sex}}$ = 47 for Study 2). Second, the results did replicate in that the significant results from Studies 1 and 2 remained significant when we changed the classification method. This also means we were not able to find an effect of context on PDA in Study 2, regardless of the method used to classify the two groups. We also transferred the original grouping method from Studies 1 and 2 to Study 3 (LGBTQ*/Hetero Cis instead of same-sex/different-sex). Again, the results remained the same with one exception: Whereas the type of relationship (with $n_{\text{same-sex}}$ = 80 and $n_{\text{different-sex}}$ = 188) had a significant main effect on displays of affection, Pillai's trace $F(2, 265)$ = 8.04, $p < .001$, sexual and gender identity ($n_{\text{LGBTQ*}}$ = 139 and $n_{\text{Hetero/Cis}}$ = 129) did not, Pillai's trace $F(2, 265)$ = 1.88, $p = .154$. Meanwhile, the main effect of context on displays of affection, Pillai's trace $F(4, 263)$ = 144.15, $p < .001$, and the interaction between context and sexual and gender identity, Pillai's trace $F(4, 26)$ = 4.29, $p = .002$, remained significant.

## Future research

In future research, a study could investigate the perception of LGBTQ* minority stress and PDA while the participants are actually present in the respective context. Alternatively, the

descriptions of the chosen contexts could be augmented with more information and pictures to ensure that participants think of the same places while answering the questions hypothetically. Also, current research has begun to explore couple-level minority stress by using a dyadic sample of couples instead of individual participants [21, 44]. This change may help disentangle the effects of individual and couple-level minority stress and provide an even better understanding of the matter of open displays of relationship behavior. For a scale that specifically measures couple-level minority stress, see Neilands et al. [45].

Second, whereas we presumed that only our minority group (LGBTQ*/same-sex) is negatively affected by LGBTQ* minority stress, participants in the other group may also be affected by minority stress if they are visible members of other minority groups. In our studies, Hetero/Cis participants had a strong preference for the city center when we asked them about their PDA. Some of the reasons could be found in the qualitative analysis of Study 3, such as being anonymous and not having to fear being perceived as unprofessional. This could be further investigated in future research to achieve a better understanding of the social component of relationship and dating behavior (e.g., by specifically examining the role of professionalism).

Third, the qualitative results from Study 3 also help elucidate why participants perceive greater minority stress in the city center compared with on campus. The themes we identified, such as "protective" and "harmful" factors, could be measured quantitatively and tested as mediators in a subsequent study. This will allow researchers to identify more tangible contextual characteristics (e.g., the presence of rainbow flags) that could explain differences between these two contexts in our studies. In this way, results can be generalized to or used as a foundation from which to generate hypotheses about what would happen in other contexts (e.g., a city vs. a rural town).

## Conclusion

The present research shows that a harmful social context with minority stress reduces LGBTQ* couples' enjoyment of public displays of affection. It also shows that a social context with less minority stress is perceived as more protective to LGBTQ* couples' and may therefore buffer the negative effect a harmful context may have on their displays of affection. Thus, a protective social context can be a powerful tool for promoting healthy LGBTQ* relationship behavior.

## Supporting information

**S1 Materials. Exploratory measures in Study 1.**
(DOCX)

**S2 Materials. Exploratory measures in Study 2.**
(DOCX)

**S3 Materials. Exploratory measures in Study 3.**
(DOCX)

**S1 Table. Locations of participants in Study 3.**
(DOCX)

## Acknowledgments

We would like to thank Yasmin Uyar, Dr. Eric Igou, Dr. Ronni Greenwood, Madison Gray, and Prof. Dr. Malte Friese for their generous support.

## Author Contributions

**Conceptualization:** Michelle Stammwitz, Janet Wessler.

**Data curation:** Michelle Stammwitz.

**Formal analysis:** Michelle Stammwitz, Janet Wessler.

**Investigation:** Michelle Stammwitz.

**Methodology:** Michelle Stammwitz, Janet Wessler.

**Supervision:** Janet Wessler.

**Writing – original draft:** Michelle Stammwitz.

**Writing – review & editing:** Michelle Stammwitz, Janet Wessler.

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
