## [Decision Letter · Decision Letter 0]

18 May 2021

PONE-D-21-11276

A Harmful vs. Protective Public Context with Higher Minority Stress for LGBTQ* Couples Decreases Enjoying Public Displays of Affection

PLOS ONE

Dear Dr. Hess,

Thank you for submitting your manuscript to PLOS ONE. After careful consideration, we feel that it has merit but does not fully meet PLOS ONE’s publication criteria as it currently stands. Therefore, we invite you to submit a revised version of the manuscript that addresses the points raised during the review process.

**Given the rich review offered by this one Reviewer and the difficulty of finding a second Reviewer, I believe that authors can take advantage of the advice offered here to make their article suitable for publication.**

We look forward to receiving your revised manuscript.

Kind regards,

Stefano Federici, Ph.D.

Academic Editor

PLOS ONE

Additional Editor Comments:

Given the rich review offered by this one Reviewer and the difficulty of finding a second Reviewer, I believe that authors can take advantage of the advice offered here to make their article suitable for publication.

Journal Requirements:

2. Please amend your manuscript to include your abstract after the title page.

Reviewers' comments:

Reviewer's Responses to Questions

**Comments to the Author**

1. Is the manuscript technically sound, and do the data support the conclusions?

Reviewer #1: Partly

2. Has the statistical analysis been performed appropriately and rigorously? 

Reviewer #1: Yes

3. Have the authors made all data underlying the findings in their manuscript fully available?

Reviewer #1: Yes

4. Is the manuscript presented in an intelligible fashion and written in standard English?

Reviewer #1: Yes

5. Review Comments to the Author

Reviewer #1: Comments to the authors:

The paper aims to investigate whether minority stress (e.g., perceived discrimination) and physical displays of affection (PDA) vary across contexts (i.e., campus, city center, and private home – Study 3). It reports three studies conducted in Ireland and Germany among both cis-heterosexual and LGBTIQ+ university students. Notably, the last study relies on both quantitative and qualitative data.

Overall, I very much enjoyed reading this paper. The manuscript is clear, generally well written, and addresses an interesting topic. I particularly appreciate the use of a mixed-method approach in Study 3, which is a great strength of this manuscript. The findings of the qualitative analyses show a very interesting opposing/conflictual effect of context on PDA. I believe this finding is of particular relevance to the literature. I also appreciate that the authors focus on the context, which is often overlooked in the literature.

Next, I list the issues I identified.

Major:

The manuscript focuses on the concept of ‘minority stress (Meyer, 2003)’. This concept is, however, underdeveloped in the introduction. I would like the authors to develop this section further (e.g., by also referring to Hatzenbuehler research). This is particularly important because the authors mainly focus on one dimension of the model (i.e., distal stressors) without considering other forms of stressors (i.e., proximal stressors).

The method used to group participants differs between Studies 1-2 (i.e., LGBTIQ+ vs. cis heterosexual) and Study 3 (same-sex vs. other sex partners). I do understand the author's reasoning, but I see it as problematic for different reasons. First, the authors mention in Study 3 “An important change compared to Study 1 and 2 was that each participant was asked to explicitly state if their (hypothetical) partner was of the same or the different sex” (p.22); however, on page 11 (Study 1) it is mentioned that “Participants indicated which gender male or female, their (hypothetical) partner has (would have).” Please clarify whether this question was also asked in Studies 1 and 2. If so, please indicate whether the results of Studies 1-2 replicate using the Study 3 classification method. I would like to know whether part of (as not all items were asked to all participants) the results of Study 3 replicate using Studies 1-2 classification method.

Second, I think that the method used in Study 3 is quite problematic for bisexual/pansexual people. In general, considering the high levels of identity denial that bisexual/pansexual people have to face (see also Maimon et al., 2019), asking them to clearly state the gender of their imagined partner (if this was the case) is a bit problematic. While this concern cannot be addressed at this stage of the research, I would like the author to not refer to LGBTIQ+ vs. heteronormative (or cis-heterosexual) relationships in the results of Study 3 (as some bisexual people were classified in the heterosexual group). As stated by the author, “this way of assigning does not conform with the actual terminology of LGBTQ* which includes bisexuals with no regard to their partner’s gender” (p, 19). Please instead refer to different vs. same-sex partners. Please also indicate how you classified the 25 participants who did not identify as male or female.

Minor:

Please indicate the year of data collection and (if possible) the cities in which Studies 1&2 data were collected. Given that this article focuses on the ‘context,’ I think this information is of particular importance.

I would prefer to see the descriptive tables after the measure section. Please also add the p-values (for the correlations) in the tables.

The term ‘sexual identity’ is used to refer to the difference between the two groups (LGBTQ+ vs. cis heterosexual). However, heterosexual trans people in Studies 1 and 2 were included in the LGBTQ+ sample, right? This is, thus, not consistent with the terminology (since their sexual identity is ‘heterosexual’). Please address this concern (e.g., by referring to sexual and gender identity).

Study 3 is presented as a higher-powered study. Please revise this claim as the LBGTQ+ sample is not larger than in Study 2. Further, the design is quite unbalanced, which might impact some of the interactions.

On page 19, it is stated that “the survey was completed by N = 268 university students at more than 30 different German universities with NLGBTQ* = 80 and NHetero/Cis = 129”. The numbers don’t add up. Please clarify why (I might have missed something).

The result section of Study 3 (quantitative part in particular) is quite long; maybe there is an option to condense part of it (e.g., by referring more to the figure/Table).

In the qualitative part of Study 3, it is mentioned that “some participants answered the initial question on why their PDA was more frequent and enjoyable in one context by explaining why their PDA was less frequent and enjoyable in the other context.”(p.29). This also indicates a possible order effect. I would be curious to see whether the quantitative results were impacted by the question order (e.g., whether one context was asked before another).

On page 30, it is stated that “Moreover, while a context with increased minority stress does not seem to reduce the frequency in which LGBTQ*couples show affection (Study 3), it does impair their emotional affect during these affection displays…”. I might be mistaken, but is it not Study 2?

I sometimes felt like the discussion could be more integrated with the current literature (and what future research can learn from your findings).

Reference:

Maimon, M. R., Sanchez, D. T., Albuja, A. F., & Howansky, K. (2019). Bisexual identity denial and health: Exploring the role of societal meta-perceptions and belonging threats among bisexual adults. Self and Identity, 1-13.

6. PLOS authors have the option to publish the peer review history of their article (what does this mean?). If published, this will include your full peer review and any attached files.

Reviewer #1: No

---

## [Author Response · Author response to Decision Letter 0]

22 Sep 2021

We thank the reviewer for the time and thought they have invested in our work. The feedback we received helped us a lot to craft a much stronger contribution. For a detailed response to the reviewer's comments see the file 'Revision Notes'.

---

## [Decision Letter · Decision Letter 1]

5 Oct 2021

PONE-D-21-11276R1A public context with higher minority stress for LGBTQ* couples decreases the enjoyment of public displays of affectionPLOS ONE

Dear Dr. Wessler,

Thank you for submitting your manuscript to PLOS ONE. After careful consideration, we feel that it has merit but does not fully meet PLOS ONE’s publication criteria as it currently stands. Therefore, we invite you to submit a revised version of the manuscript that addresses the points raised during the review process.

**A little more effort to address some minor Reviewer's comments and then the decision will be made by the Academic Editor without following through with a third review.**

We look forward to receiving your revised manuscript.

Kind regards,

Stefano Federici, Ph.D.

Academic Editor

PLOS ONE

Journal Requirements:

Additional Editor Comments (if provided):

A little more effort to address some minor Reviewer's comments and then the decision will be made by the Academic Editor without following through with a third review..

Reviewers' comments:

Reviewer's Responses to Questions

**Comments to the Author**

1. If the authors have adequately addressed your comments raised in a previous round of review and you feel that this manuscript is now acceptable for publication, you may indicate that here to bypass the “Comments to the Author” section, enter your conflict of interest statement in the “Confidential to Editor” section, and submit your "Accept" recommendation.

Reviewer #1: (No Response)

2. Is the manuscript technically sound, and do the data support the conclusions?

Reviewer #1: Yes

3. Has the statistical analysis been performed appropriately and rigorously? 

Reviewer #1: Yes

4. Have the authors made all data underlying the findings in their manuscript fully available?

Reviewer #1: (No Response)

5. Is the manuscript presented in an intelligible fashion and written in standard English?

Reviewer #1: Yes

6. Review Comments to the Author

Reviewer #1: I appreciate that the authors addressed my comments thoroughly. I am, in general, very happy with the revisions. I, however, still have a few minor concerns:

1. The authors did a great job in developing the ‘minority stress model’ in the introduction. On a minor note, the authors might consider splitting the new paragraph on p.4 into two (e.g., splitting at lines 36-37).

2. The description of the method used to group participants is much clearer now. The authors mention that the question in Studies 1-2 read: “What gender does/would your partner have? [Female/Male/Other]”. However, in the text, it is mentioned that “Participants indicated which gender male or female, their (hypothetical) partner has.” Please revise the text to mention that participants could also indicate that they had a partner who did not fall into the gender binarity (i.e., the ‘other’ response category).

3. I was also happy to see the new analyses using a different method to group participants in the discussion section. If the authors are short on the word limit (or want to shorten their discussion section), they might consider moving these findings to the supplementary material and mentioning them in the main text (or in a footnote).

4. Finally, I thank the authors for mentioning the limitations of the study regarding bi- and pansexual participants. In general, I like this addition. However, I still find the sentence “because our research focused on distal stressors, which are more pronounced when people are perceived as LGBTQ* by other […]” problematic. Indeed, if people are genuinely perceived as LGBTQ*, then this should not be depending on the sex of their partner (i.e., bisexual people should be considered LGBTQ* regardless of the sex of their partner). I would therefore suggest revising this sentence to, for instance: “because our research focused on distal stressors, which are more pronounced when people are with a same-sex vs. different-sex partner […]”.

7. PLOS authors have the option to publish the peer review history of their article (what does this mean?). If published, this will include your full peer review and any attached files.

Reviewer #1: No

---

## [Author Response · Author response to Decision Letter 1]

12 Oct 2021

Reviewer #1

I appreciate that the authors addressed my comments thoroughly. I am, in general, very happy with the revisions. I, however, still have a few minor concerns:

→ Thank you very much for this positive feedback. We are happy that we were able to address your comments in detail.

1. The authors did a great job in developing the ‘minority stress model’ in the introduction. On a minor note, the authors might consider splitting the new paragraph on p.4 into two (e.g., splitting at lines 36-37).

→ We now splitted the paragraph on page 4 at lines 32-33, in order to differentiate between the paragraph dealing with distal and proximal stressors from the paragraph dealing with the negative consequences of experiencing minority stress. 

2. The description of the method used to group participants is much clearer now. The authors mention that the question in Studies 1-2 read: “What gender does/would your partner have? [Fem ale/Male/Other]”. However, in the text, it is mentioned that “Participants indicated which gender male or female, their (hypothetical) partner has.” Please revise the text to mention that participants could also indicate that they had a partner who did not fall into the gender binarity (i.e., the ‘other’ response category).

→ On page12 ll. 235-236, we now included the other category in this information: 

„Participants indicated the gender of their (hypothetical) partner (female/male/other).“

3. I was also happy to see the new analyses using a different method to group participants in the discussion section. If the authors are short on the word limit (or want to shorten their discussion section), they might consider moving these findings to the supplementary material and mentioning them in the main text (or in a footnote).

→ We are glad that the reviewer liked our addition about the grouping information. We decided to retain this addition in the limitation section, because footnotes are not an formal option at the journal and we think the information is worth to be mentioned in the main manuscript.

4. Finally, I thank the authors for mentioning the limitations of the study regarding bi- and pansexual participants. In general, I like this addition. However, I still find the sentence “because our research focused on distal stressors, which are more pronounced when people are perceived as LGBTQ* by other […]” problematic. Indeed, if people are genuinely perceived as LGBTQ*, then this should not be depending on the sex of their partner (i.e., bisexual people should be considered LGBTQ* regardless of the sex of their partner). I would therefore suggest revising this sentence to, for instance: “because our research focused on distal stressors, which are more pronounced when people are with a same-sex vs. different-sex partner […]”.

→ We now changed the sentence on p. 22 ll. 467-468 to:

„Because our research focused on distal stressors, which are more pronounced when people are with a same-sex vs. different-sex partner,...“

---

## [Editor Report · Decision Letter 2]

13 Oct 2021

A public context with higher minority stress for LGBTQ* couples decreases the enjoyment of public displays of affection

PONE-D-21-11276R2

Dear Dr. Wessler,

We’re pleased to inform you that your manuscript has been judged scientifically suitable for publication and will be formally accepted for publication once it meets all outstanding technical requirements.

Kind regards,

Stefano Federici, Ph.D.

Academic Editor

PLOS ONE
---

## [Editor Report · Acceptance letter]

27 Oct 2021

PONE-D-21-11276R2 

A public context with higher minority stress for LGBTQ* couples decreases the enjoyment of public displays of affection 

Dear Dr. Wessler:

I'm pleased to inform you that your manuscript has been deemed suitable for publication in PLOS ONE. Congratulations! Your manuscript is now with our production department. 

Kind regards, 

on behalf of

Prof. Stefano Federici 

Academic Editor

PLOS ONE